ecology/ecosystems/environmental science

soil fauna, grubbing, invasion, wild boar, ecosystem services, ecosystem functions

**Author for correspondence:**
Nadia I. Maaroufi
e-mail: nadia.maaroufi@slu.se

# Northward range expansion of rooting ungulates decreases detritivore and predatory mite abundances in boreal forests

Nadia I. Maaroufi[1,2,3], Astrid R. Taylor[1],
Roswitha B. Ehnes[1], Henrik Andrén[4], Petter Kjellander[4],
Christer Björkman[1], Thomas Kätterer[1] and
Maartje J. Klapwijk[1]

[1]Department of Ecology, and [2]Department of Forest Mycology and Plant Pathology, Swedish University of Agricultural Sciences (SLU), 756 51 Uppsala, Sweden
[3]Institute of Plant Sciences, University of Bern, 3013 Bern, Switzerland
[4]Department of Ecology, Swedish University of Agricultural Sciences (SLU), Grimsö Wildlife Research Station, 730 91 Riddarhyttan, Sweden

NIM, 0000-0002-8028-1409; HA, 0000-0002-5616-2426

In the last few decades wild boar populations have expanded northwards, colonizing boreal forests. The soil disturbances caused by wild boar rooting may have an impact on soil organisms that play a key role in organic matter turnover. However, the impact of wild boar colonization on boreal forest ecosystems and soil organisms remains largely unknown. We investigated the effect of natural and simulated rooting on decomposer and predatory soil mites (total, adult and juvenile abundances; and adult–juvenile proportion). Our simulated rooting experiment aimed to disentangle the effects of (i) bioturbation due to soil mixing and (ii) removing organic material (wild boar food resources) on soil mites. Our results showed a decline in the abundance of adult soil mites in response to both natural and artificial rooting, while juvenile abundance and the relative proportion of adults and juveniles were not affected. The expansion of wild boar northwards and into new habitats has negative effects on soil decomposer abundances in boreal forests which may cascade through the soil food web ultimately affecting ecosystem processes. Our study also suggests that a combined use of natural and controlled experimental approaches is the way forward to reveal any subtle interaction between aboveground and belowground organisms and the ecosystem functions they drive.

# 1. Introduction

Wild boar (*Sus scrofa*) is among the most widely distributed ungulates worldwide, both within and beyond its Eurasian–North African native range [1]. In recent decades, the European wild boar population has increased and its Scandinavian population is currently expanding northwards, colonizing boreal forests dominated by coniferous trees [2,3].

In the last few decades, several studies have provided evidence that ungulates have an extensive impact on a wide range of ecosystems such as grasslands, alpine meadows and deciduous forests when foraging for aboveground plant biomass [4–6]. For instance, in grasslands, the presence of large ungulates has been shown to modify the strength of the interactions between above- and belowground invertebrates, and between soil detritivores and nutrient cycling [7,8]. Selective foraging and foraging frequency alter the quantity and quality of litter entering soil food webs [9]. However, some ungulates are omnivorous, such as wild boar, and consume belowground plant organs, fungi, invertebrates and soil organic matter (i.e. rooting, grubbing) [10]. Wild boar rooting behaviour consists of bioturbating the soil by disturbing the understory layer vegetation, pushing litter away or burying this organic material on the same spot [3,11,12]. Rooting activities may impact soil fauna by altering the biomass and the activity of primary producers and consumers, e.g. effects on belowground plant roots and their associated symbionts which feed back on plant community structure and plant litter inputs.

A stronger focus on soil fauna in research has led to the insight that soil organisms drive key ecosystem functions, e.g. plant nutrient availability, organic matter turnover and soil carbon storage [7,13,14]. In particular, soil mites are known to play a role in organic matter turnover and are contributing up to 45% of the total soil fauna respiration [15,16]. However, studies on soil organisms and their response to wild boar rooting are scarce [17,18]. The studies addressing effects of wild boar rooting on soil biota applied simulated rooting treatments by disrupting soil horizons [19,20]. These experiments successfully simulated one part of the rooting behaviour but did not assess the effect of actual removal of biomass from the soil. Hence, in this study we focus on the combined effect of rooting and organic matter removal to make a first assessment of the potential combined impact on soil stoichiometry and detritivore and predatory mite abundances. We divided the mite community into these two different groups to assess the effects on two different functional groups.

In this study, we explored the effect of natural and simulated rooting on two groups of soil mites that differ in their feeding behaviour, detritivore oribatid mites and predatory mesostigmatid mites. Our aims were to (i) assess the effect of natural wild boar rooting behaviour on the soil mite community, (ii) validate our method for simulated rooting by comparing its effects with that of natural rooting, and finally (iii) disentangle effects of bioturbation (i.e. soil disturbance) and removal of organic material as two main impacts of wild boar rooting behaviour on soil mites.

# 2. Material and methods

## 2.1. Study site

The study was performed at Bornjsön Vatten Reservat, a forested area south of Stockholm (www. stockholmvattenochavfall.se/en/, 59°44′ N, 17°47′ E), in the hemi-boreal zone of eastern Sweden [21]. In 2017, six forest sites with similar vegetation type were selected consisting of Norway spruce (*Picea abies* (L.) Karst) with occasional Scots pine (*Pinus sylvestris* L.), oak (*Quercus robur*) and birch (*Betula pendula* and *B. pubescens*). The understory layer is dominated by the ericaceous dwarf-shrub *Vaccinium myrtillus* L. and the grass species *Deschampsia flexuosa* (L.) Trin. The bottom layer moss-mat consists of *Hylocomium splendens* (Hedw.) B.S.G, *Pleurozium schreberi* (Bird), and *Ptilium crista-castrensis* (Hedw.). Bedrock at the site is mainly gneissic and soils are Dystric to Eutric Cambisols developed from glacial till and clay.

## 2.2. Site selection and exclosure design

In August 2017, six sites (*ca* 0.1 ha) with similar tree species composition and understory vegetation were selected (see the Study site section). All sites were located at least 1 km apart from each other. In summer 2017, we established a simulated rooting experiment to investigate the impact of wild boar rooting frequency on soil mites in boreal forests. At each site, areas with no apparent rooting activity were

chosen. These sites had intact plant understory layer (i.e. no sign of trampling or other damages) and typical plant species composition as the rest of the surrounding forest. We erected wild boar exclosures by creating fenced areas to control the level of rooting damage and thus reduce the risk of additional disturbances caused by wild boar or other large mammals (e.g. row deer). Within each site, an area of 36 m² was fenced using 1.50 m high galvanized mesh (10 cm size) and a line of barbed wire applied along the bottom fence perimeter to avoid wild boar lifting up the fence [19,22].

## 2.3. Artificial wild boar rooting experiment

Wild boar rooting activity consists of disturbing the understory layer vegetation, pushing litter and the bryophyte layer away or burying this organic material on the same spot. They consume seedlings, plant shoots, fruits, seeds, gastropods, batrachians and arthropods, which makes wild boar an omnivorous ungulate species [10,23,24]. Further, wild boar feed on soil organisms by excavating soil, and they consume roots, bulbs, hyphal networks, arthropod larvae and other soil fauna such as earthworms [20,25]. Rooting areas of wild boar are highly variable in size and can range from 1 m² to 1 ha [3,20]. In addition, rooting depth is highly variable. Previous studies reported an average depth between 5 and 15 cm, but wild boar can root as deep as 45 cm [3,12].

In September 2017, within each exclosure, five rectangular plots (5.25 m², 3.5 m × 1.5 m) were randomly assigned to one of the five treatments. Each plot was surrounded by a buffer zone of greater than or equal to 30 cm. The control (C) was not bioturbated nor was biomass removed. The rooting treatments consisted of a low level of bioturbation (LB; soil bioturbated once in September 2017), a high level of bioturbation (HB; in September and October 2017, three weeks apart), a low level of bioturbation and removal of organic material (LB + Re; complete rooting done once) and a high level of bioturbation and removal of organic material (HB + Re; complete rooting done twice, three weeks apart). Thus, our design consisted of two treatments, bioturbation (B) and removal (Re) with two levels (low or high) and a control. All treatments were repeated in each of the six exclosures (block). The experiment set-up allowed two types of comparisons: (i) all treatment combinations with the control and (ii) a full factorial design with two levels within each two treatments, bioturbation (low or high) and removal (no biomass removal or biomass removal). We performed rooting treatments in September–October when wild boar rooting activity is occurring in our forest system, which coincides with the increase of sporocarp production, freshly fallen leaf litter and seeds, as well as the presence of *V. myrtillus* shoots and berries.

The bioturbation treatment was performed using a modified claw soil mixer with tubular handle (Freund Victoria, Germany) and a 50 cm × 50 cm metal frame. The metal frame was applied to the ground and moved following parallel rows to get a homogeneous bioturbation effort throughout the plot. The soil mixer was forced into the ground at 10 cm depth and a 180° rotation movement was applied before taking the soil mixer out of the ground. The removal treatment consisted of removing any organic material and organisms known to be consumed by wild boar (see the beginning of this section). In practice, the metal frame was put into the ground, and all potential wild boar food resources were removed with one hand during 1 min per frame to get the same removal effort throughout the plots.

## 2.4. Natural wild boar rooting

In September 2018, we set up two additional plots in close vicinity of the existing exclosures (i.e. 12 plots in total, *ca* 100 m² each), the selected plots being covered by intact ground vegetation indicating the absence of recent rooting activities. In October 2018, we induced natural wild boar rooting within each plot by spraying tar on tree trunks to attract wild boar (Auson AB, Sweden). The latter experiment (referred to as natural rooting experiment) consisted of two treatments: natural rooting sites (NR) and adjacent non-rooted sites (NC; natural control). The non-rooted plots were paired with the adjacent wild boar rooted plots, and consisted of areas with no sign of trampling and rooting.

## 2.5. Soil mite community

Four weeks after simulated rooting in the exclosures (October 2017) and after spraying tar (October 2018), two cylindrical soil core samples (5.5 cm diameter, 10 cm depth) were randomly taken from each treatment, transferred to sealed plastic containers and placed in Tullgren extractors for five days to extract the micro-arthropods [26]. All extracted animals were stored in vials containing 80% ethanol

for preservation. Mites were identified to sub-order levels: Oribatida (fungivores, detritivores) and Mesostigmata (predators) [27,28]. Further, we divided the Mesostigmata and Oribatida taxa into two growth stages, juveniles and adults. Abundance estimates from each plot were derived from the average of two subsamples within each plot. We excluded one of the six natural rooting sites within the experiment from our sampling because it occurred that wild boar did not come rooting in one of the sites. Thus, five sites were used for this study making a total of 15 and 10 units for the artificial and natural wild boar rooting, respectively.

## 2.6. Soil stoichiometry and properties

In October 2017 and 2018 (see previous section), two cylindrical soil core samples (2.5 cm diameter, 10 cm depth) were randomly taken from each treatment and transferred to sealed plastic bags. The two cores were pooled to obtain one composite sample per plot. The samples were passed through a 4 mm sieve to remove roots, stones, and coarse materials, then homogenized. A subsample was oven-dried at 60°C for 48 h. This subsample was ground for chemical characterization using a ball mill (Retsch MM301; Haan, Germany). Total carbon (C) and nitrogen (N) were determined by dry combustion (Flash EA 2000; Thermo Fisher Scientific, Germany) while total phosphorus (P) was measured by Kjeldahl acid digestion (Auto Analyzer III Spectrophotometer; Omniprocess, Germany). Another subsample was taken for soil water content analysis by oven drying at 105°C for 48 h [29]. Soil organic matter content was taken to measure the soil organic matter content determined by loss on ignition at 550°C for 6 h [27].

## 2.7. Statistical analyses

We performed three separate analyses, to test whether Mesostigmata and Oribatida communities (i.e. total, adult and juvenile abundance, adult–juvenile proportions) (i) differed between natural wild boar rooted areas and adjacent non-rooted area, (ii) differed between artificial rooting treatments and the control, and (iii) differed in the bioturbation and removal treatments when excluding the control from the model. The three analyses were conducted using generalized linear mixed models (GLMMs) with the function glmer from the lme4 R package [30]. We used the Poisson error distribution for count data (abundance) and the binomial distribution for proportion data (here the adult–juvenile proportional presence) [31]. When necessary, we accounted for over-dispersion by including sample identity as a random factor [32]. Block was included as a random factor to account for site differences.

(i) We first conducted a GLMM comparing natural wild boar rooting with corresponding non-rooted patches including block and sample identity as random factors. (ii) We then performed a GLMM to compare the four treatment combinations against the control with block and sample identity included as random factors. When significant differences between treatments were detected ($\alpha \leq 0.05$), *post hoc* pairwise comparisons were conducted using the Tukey test. (iii) Lastly, we performed a GLMM using bioturbation (two levels) and removal (two levels) as fixed factors, and the interactions between rooting and removal. Block and sample identity were included as random factors.

The change with rooting treatment in total C, N, P and associated ratios as well as soil moisture and organic matter content were analysed using linear mixed models (LMMs) with the function lmer from the lme4 R package [30]. When the assumption of normality was not met GLMM models were used with the Poisson error distribution for count data (abundance) [31]. Block was included as a random factor to account for site differences.

All statistical analyses were performed using R v. 4.0.3 (R Core Team 2020).

# 3. Results

## 3.1. Soil stoichiometry and parameters

### 3.1.1. Natural rooting

The soil C : N ratio significantly declined by 13% in response to natural wild boar rooting (Control 23.57 ± 1.83, Rooting 20.18 ± 0.93, $p = 0.005$; table 1). Natural rooting caused a nearly significant increase of C : P ratio (Control 252.25 ± 68.33, Rooting 271.70 ± 126.62, $p = 0.056$). Soil moisture (Control 27.20 ± 4.53, Rooting 23.78 ± 3.22, $p = 0.017$) and organic matter content (Control 26.30 ± 8.42, Rooting 18.49 ± 3.63, $p = 0.011$) were significantly reduced under natural rooting compared to unrooted patches

**Table 1.** Mean ± s.e. of % carbon, nitrogen, phosphorus and associated ratios for forest soil in response to wild boar rooting treatments: unrooted control, wild boar rooting. The $\chi^2$- and $p$-values were derived from ANOVAs for generalized linear mixed models ($\chi^2_{1,7}$). Values in bold indicate statistical significances at $p < 0.05$. Asterisk indicates marginally significant difference ($0.05 \leq p < 0.1$).

| | treatments | | | |
| --- | --- | --- | --- | --- |
| | control | wild boar rooting | $\chi^2$ | $p$-value |
| *soil stoichiometry* | | | | |
| %C | 13.54 ± 4.22 | 13.93 ± 4.75 | 0.01 | 0.932 |
| %N | 0.54 ± 0.14 | 0.67 ± 0.21 | 0.20 | 0.657 |
| %P | 0.05 ± 0.01 | 0.06 ± 0.01 | 1.88 | 0.169 |
| C : N | 23.57 ± 1.83 | 20.18 ± 0.93 | 7.92 | **0.005** |
| C : P | 252.25 ± 68.33 | 271.70 ± 126.62 | 3.66 | 0.056* |
| N : P | 10.14 ± 2.27 | 12.98 ± 5.66 | 1.68 | 0.195 |
| *soil parameters* | | | | |
| % moisture | 27.20 ± 4.53 | 23.78 ± 3.22 | 5.74 | **0.017** |
| % organic matter content | 26.30 ± 8.42 | 18.49 ± 3.63 | 6.38 | **0.011** |

by 12.6% and 29.7%, respectively. Our data did not reveal any significant differences in total soil C, N, P and N : P ratio in response to natural wild boar rooting.

### 3.1.2. Simulated rooting

Soil stoichiometry and parameters did not significantly respond to simulated wild boar rooting. We only found a non-significant negative trend in the organic matter content in response to simulated rooting treatments (table 2; $p = 0.09$).

## 3.2. Soil mite community

### 3.2.1. Natural rooting

Total and adult abundances of Oribatida were significantly lower under natural rooting compared to unrooted patches by 75.4% (Unrooted 121.64 ± 54.01, Rooted 29.99 ± 9.64; $p < 0.0001$) and 72.8% (Unrooted 83.8 ± 28.75, Rooted 22.8 ± 8.11, $p < 0.0001$), respectively (figure 1a,b; electronic supplementary material, table S1). The number of oribatid juveniles (figure 1b; $p = 0.371$) as well as the relative abundance of juveniles in the sample (figure 1c; $p = 0.163$) were not significantly different between rooted and unrooted areas. Total and adult abundances of Mesostigmata were lower under natural rooting compared to unrooted ones by 64.8% (Unrooted 12.84 ± 4.65, Rooted 4.52 ± 1.05, $p = 0.019$) and 66.1% (Unrooted 11.20 ± 3.98, Rooted 3.80 ± 1.02, $p = 0.034$), respectively (figure 1d,e). The number of mesostigmatid juveniles (figure 1e; $p = 0.147$) as well as the relative abundance of juveniles in the sample (figure 1f; $p = 0.818$) were not significantly different between rooted and unrooted areas.

### 3.2.2. Simulated rooting

Total and adult Oribatida abundances also decreased in response to simulated rooting treatments (figure 2a,b). The largest decline was observed in the low bioturbation and removal treatment (56.5%, LB + Re 115.12 ± 23.53) and the smallest decline was observed in the high bioturbation and removal treatment (32.3%, HB + Re 148.34 ± 58.83, figure 2a; electronic supplementary material, table S1). Adult Oribatida abundances followed the same trend as the total abundance (figure 2b; electronic supplementary material, table S1). The oribatid adult–juvenile relative abundance was not affected by simulating rooting (figure 2c; electronic supplementary material, table S2). For Mesostigmata, we found that rooting treatments caused a nearly significant decline of total and adult abundances (figure 2d,e; $p < 0.1$). The mesostigmatid adult–juvenile relative abundance was not affected by

**Table 2.** Mean ± s.e. of % carbon, nitrogen, phosphorus and associated ratios for forest soil in response to simulated wild boar rooting treatments: control, low bioturbation, low bioturbation + removal, high bioturbation, high bioturbation + removal, high bioturbation + removal. The $\chi^2$- and p-values ($\chi^2_{(4,18)}$) were derived from ANOVAs for linear mixed models or generalized linear mixed models (‡). Asterisk indicates marginally significant difference ($0.05 \leq p < 0.1$).

| | treatments | | | | | $\chi^2$ | p-value |
|---|---|---|---|---|---|---|---|
| | control | low bioturbation | low bioturbation + removal | high bioturbation | high bioturbation + removal | | |
| **soil stoichiometry** | | | | | | | |
| %C | 18.78 ± 6.52 | 14.83 ± 3.97 | 15.46 ± 5.82 | 17.76 ± 6.60 | 17.63 ± 6.46 | 5.09 | 0.279 |
| %N | 0.69 ± 0.18 | 0.58 ± 0.12 | 0.58 ± 0.15 | 0.66 ± 0.19 | 0.66 ± 0.19 | 8.26 | 0.083‡* |
| %P | 0.11 ± 0.01 | 0.10 ± 0.01 | 0.10 ± 0.01 | 0.10 ± 0.00 | 0.11 ± 0.01 | 0.28 | 0.991‡ |
| C : N | 24.69 ± 2.39 | 24.74 ± 2.14 | 24.22 ± 2.61 | 24.36 ± 2.45 | 24.95 ± 2.34 | 0.13 | 0.998‡ |
| C : P | 171.27 ± 50.61 | 154.37 ± 46.80 | 153.22 ± 56.99 | 177.48 ± 67.45 | 165.93 ± 63.14 | 3.49 | 0.479‡ |
| N : P | 6.41 ± 1.41 | 5.93 ± 1.42 | 5.76 ± 1.55 | 6.59 ± 1.94 | 6.08 ± 1.84 | 1.43 | 0.839 |
| **soil parameters** | | | | | | | |
| % moisture | 30.80 ± 4.97 | 30.78 ± 3.47 | 27.29 ± 3.12 | 30.09 ± 4.45 | 27.63 ± 3.14 | 1.53 | 0.820 |
| % organic matter content | 34.16 ± 11.38 | 24.25 ± 5.56 | 25.55 ± 8.92 | 26.22 ± 8.22 | 27.74 ± 9.13 | 7.96 | 0.093* |

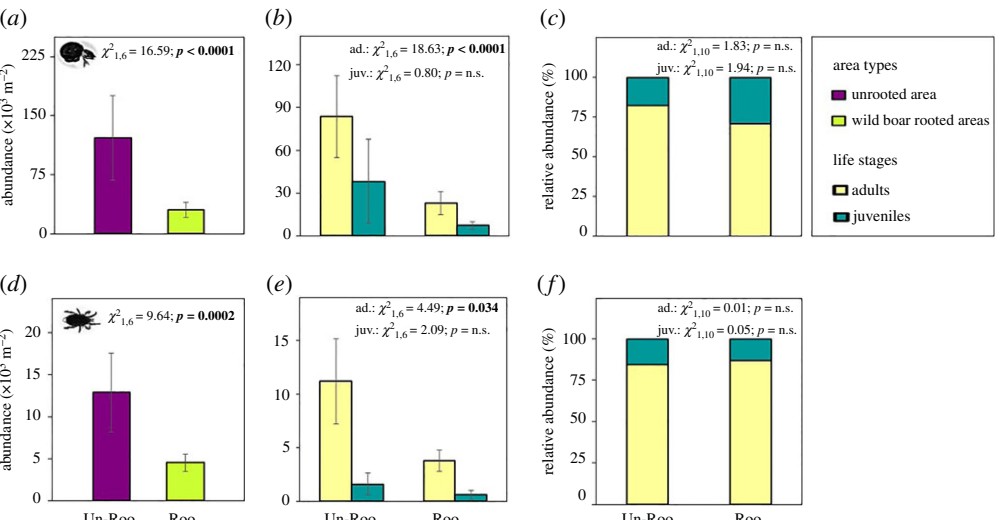

**Figure 1.** Mean (±s.e.) abundance for total (*a*), adult and juvenile (*b*), and relative abundance (*c*) Oribatida and for total (*d*), adult and juvenile (*e*), and relative abundance (*f*) Mesostigmata in response to natural wild boar rooting. *p*-values in bold indicate significant differences (*p* < 0.05). n.s. indicates non-significant.

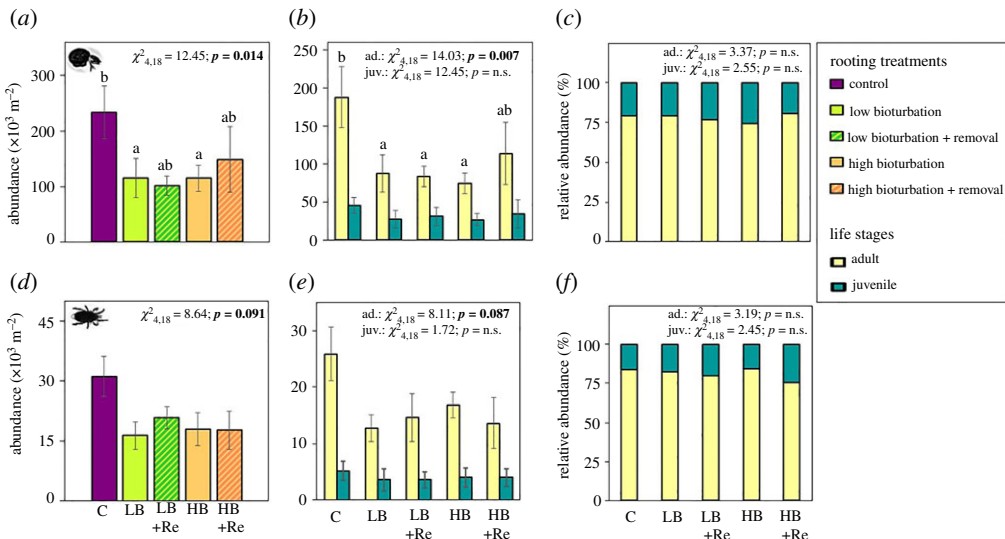

**Figure 2.** Mean (±s.e.) abundance for total (*a*), adult and juvenile (*b*), and relative abundance (*c*) Oribatida and for total (*d*), adult and juvenile (*e*), and relative abundance (*f*) Mesostigmata in response to artificial rooting treatments. Treatments consisted of control (purple, C), low bioturbation (green, LB), low bioturbation and removal (dashed green, LB + R), high bioturbation (orange, HB), high bioturbation and removal (dashed orange, HB + R). Different letters (a or b) on top of each bar indicate significant differences (*p* < 0.05) between treatments determined using Tukey *post hoc* tests. *p*-values in bold indicate significant differences (*p* < 0.05). *p*-values in italic indicate marginally significant differences (0.05 ≤ *p* < 0.1). n.s. indicates non-significant.

simulating rooting (figure 2*f*). Using a factorial ANOVA to disentangle the effects of rooting and removal of organic material or their interaction did not show any significant interactive or individual effects on soil mite abundances (table 3).

## 4. Discussion

Our main aim was to investigate how soil mites respond to wild boar rooting behaviour in boreal ecosystems. We sought to evaluate whether our method for simulated rooting showed similar effects to natural rooting on soil mites. In addition, we attempted to disentangle the effects of rooting due to

**Table 3.** Results from a two-way ANOVA for generalized linear mixed models evaluating the main and interactive effects of bioturbation (Bio) and removal (Re) on soil mite abundances.

| | estimate | s.e. | $\chi^2$ | d.f. | p-value |
|---|---|---|---|---|---|
| Oribatida | | | | | |
| *adult abundance* | | | | | |
| fixed effects | | | | | |
| intercept | 4.31 | 0.26 | | | |
| bioturbation intensity (Bio) | | | 0.06 | 1,14 | 0.803 |
| removal (Re) | | | 0.05 | 1,14 | 0.821 |
| Bio × Re | | | 0.14 | 1,14 | 0.705 |
| random effects | intercept | s.d. | | | |
| site identity | 0.08 | 0.28 | | | |
| sample identity | 0.24 | 0.49 | | | |
| *adult relative abundance* | | | | | |
| fixed effects | | | | | |
| intercept | 4.61 | 0.20 | | | |
| bioturbation intensity (Bio) | | | <0.01 | 1,14 | 0.993 |
| removal (Re) | | | 0.32 | 1,14 | 0.569 |
| Bio × Re | | | 0.01 | 1,14 | 0.930 |
| random effects | intercept | s.d. | | | |
| site identity | 0.14 | 0.38 | | | |
| sample identity | 0.23 | 0.48 | | | |

disturbance via bioturbation (soil mixing) and due to the removal of organic material consumed by wild boar on soil mite abundances. To our knowledge, we present the first experiment of this kind in the boreal region [19,22], providing a unique opportunity to investigate the effect of wild boar rooting on soil mesofauna that are important for organic matter turnover.

We showed that total Oribatida abundance declined in response to natural and artificial rooting relative to unrooted areas. The decline of the total abundance of Oribatida mites was mainly driven by the decrease in adults while the juveniles remained unaffected. These responses to rooting are similar to patterns that have been reported in studies on the effect of tillage in agricultural soils [33–36]. One mechanism that may explain the decline in soil mite abundances in both of these study types is the reduction of available habitat (i.e. the soil organic horizon), which in turn will affect accessible food resources such as litter and soil fungal biomass [37]. Further, a reduction in soil mite abundances in response to rooting could be explained by many oribatid taxa having a conservative growth strategy and a high level of habitat specialization (i.e. K-strategists) [37,38]. Further, the difference in feeding strategies between the adult and juvenile could explain the observed dissimilarity in response to the treatments. Juveniles feed mainly on microorganisms, e.g. bacteria, protists, yeasts, spores and nematodes, which likely are less affected by rooting, while adults rely primarily on organic matter of larger size, e.g. fungi, decaying plants and dead animal matter [39]. As such wild boar rooting, which results in organic and mineral soil mixing, in combination with fungal hyphal net disruption may have a stronger effect on adult than on juvenile Oribatida.

Mesostigmata abundances also declined in response to both natural and artificial rooting, again adult abundance being more affected than juvenile abundance. The abundance of Mesostigmata indicates the occurrence of their preys such as poorly sclerotized oribatid juveniles, nematodes, collembola, enchytraeids, arthropod larvae and eggs [40,41]. As these preys are primarily located in the upper soil horizons, adult Mesostigmata may have suffered from soil mixing resulting in a lower prey availability. We found marginally significant effect of the rooting treatments on Mesostigmata abundances (figure 2) which could indicate the importance of other factors, not controlled for in our experimental design. Discrepancies in soil compaction, moisture or soil physico-chemical properties

may affect Mesostigmata abundances and their response to disturbance. In our study, wild boar rooting decreases soil moisture and organic matter content as well as the soil C : N ratio which might have contributed to the decline of soil mites that we observed.

The change in soil properties and stoichiometry occurred four weeks after rooting treatments. The soil moisture content declined in response to wild boar rooting which was likely due to better soil aeration, decreasing plant cover and evapotranspiration [42]. We also found a decrease in the soil C : N ratio in response to rooting which may be likely due to more N becoming available as plant uptake declines and root debris and other organic material become available to decay [43]. These short-term effects that we observed in our experiment may have long-term effects on below- but also above-ground organisms in boreal forests. For instance, Horčičková et al. [44] showed that wild boar rooting can affect plant richness and that its effect can last more than 8 years even when rooting patches are no longer visible. Further, our experiment was conducted in autumn when precipitation events are frequent followed by snow fall and freeze–thaw events. These climatic events may affect the soil nutrient pool [45,46], as leaching may be more frequent compared to undisturbed areas.

Another aspect of rooting behaviour is the frequency and the intensity of the soil disturbance. In our experiment, we could not detect a significant effect on soil properties and stoichiometry nor on soil mite abundances in response to rooting intensity (i.e. rooting applied once or twice). This question whether the rooting intensity rate applied in our experiment was enough to produce any changes in the variables measured or whether changes were not detectable at the time of our sampling campaign. Interestingly, a study suggested that rooting effects on soil properties and stoichiometry may depend on the rooting patch size [42]. The authors found that the soil N pools were altered by rooting until a $5 \, m^3$ patch size was reached (e.g. $7 \, m^2 \times 10 \, cm$ depth) which was similar to our patch sizes (less than or equal to $5 \, m^3$, $5.25 \, m^2 \times 10 \, cm$ depth). Thus, effects of wild boar rooting on soil properties and stoichiometry may not only depend on the temporality and intensity but also the volume of soil that has been disturbed during a rooting event.

The majority of wild boar research has focused on its impact on plant communities [47,48] and on the mechanistic consequence (bioturbation) of wild boar foraging behaviour [19,20]. We showed that our short-term simulated rooting experiment had similar effects on soil mite abundance to natural rooting. As far as we know our study is the first to attempt to disentangle the importance of rooting from the removal of food resources during wild boar rooting activity. However, we did not detect any clear effects of bioturbation or removal treatment or their combination on soil mite abundances. The effect of bioturbation per se was likely so strong that it overshaded any effects from organic matter removal. As stated above, other factors not measured or controlled for in our experiment may play an important role when simulating wild boar rooting in boreal forests. We believe that testing the importance of combined factors and their interaction is the way forward to get a better understanding of the effect of wild boar rooting on soil biota and soil functioning. For a more complete understanding of the effect of wild boar on soil and ecosystem functioning, further investigations of the long-term interactions between large ungulate rooting and soil organisms are needed to disentangle possible consequences for key processes performed by soil biota.

Data accessibility. Data are available from the Dryad Digital Repository: https://doi.org/10.5061/dryad.3ffbg79jb [49].
    The data are provided in electronic supplementary material [50].
Authors' contributions. N.I.M.: conceptualization, data curation, formal analysis, investigation, methodology, visualization, writing—original draft, writing—review and editing; A.R.T.: conceptualization, resources, writing—original draft; R.B.E.: conceptualization, writing—original draft; H.A.: conceptualization, writing—original draft; P.K.: conceptualization, writing—original draft; C.B.: conceptualization, writing—original draft; T.K.: conceptualization, writing—original draft; M.J.K.: conceptualization, formal analysis, funding acquisition, resources, writing—original draft, writing—review and editing.
    All authors gave final approval for publication and agreed to be held accountable for the work performed therein.
Conflict of interest declaration. The authors declare no competing interests.
Funding. The project was supported by the Swedish University of Agricultural Sciences (SLU) for the project Unifying Ecology.
Acknowledgements. We thank Sammy Jendoubi, Ljudmila Skoglund and Kaisa Torppa for assistance with field and laboratory work. We thank Gunnar Schön at Stockholm Vatten och Avfall Bornsjön AB Bornsjö station for field assistance and giving access to the forest site.

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
