## [Peer Review File · Royal Society Open Science]

Review History

RSOS-211283.R0 (Original submission)

Review form: Reviewer 1

Is the manuscript scientifically sound in its present form?

Yes

Are the interpretations and conclusions justified by the results?

No

Is the language acceptable?

Yes

Do you have any ethical concerns with this paper?

No

Have you any concerns about statistical analyses in this paper?

No

Recommendation?

Reject

Comments to the Author(s)

I reviewed the manuscript "Northward range expansion of rooting ungulates decreases soil decomposer abundances in boreal forest" by Maaroufi et al. In this study the authors investigated the effects of natural and simulated rooting on soil mite community. Their results show a decline in the abundance of adult mites in natural and simulated rotting areas in comparison to control areas. This study is interesting in that fewer studies have looked into the effects of wild boar rooting in the soil community. However, I found it limited in the implications. It would have been nice to see changes in the food web or ecosystem processes in response to these changes as suggested in the introduction. Measuring some other response variables such as the abundance of other members of the food web, or organic matter turnover, or soil respiration, might help to show the relevance of your findings. Otherwise your results are mainly descriptive. Also, I find the title a bit misleading, as you only measured the mite community, not the soil decomposer community.

Other suggestions:

Line 195: stats are missing

Line 197: stats are missing

In general rather than describing the significance of the statistical test, I recommend describing the biological results, e.g. Soil stoichiometry and parameters did not vary with rooting treatments (STATS). Similarly, you could say: Total soil C, N, P and N:P ratio did not vary in response to natural wild boar rooting (Stats).

Line 195-201: It is a bit confusing to follow whether you are describing the results of the simulated rooting experiment or the natural rooting experiment, as you introduced them in a different order in the methods. I had to check the tables to understand which you were referring to. I suggest including a sentence at the beginning of this section saying that you found different effects on soil parameters depending on whether the disturbance was natural or simulated, and then describe each one. Alternatively you could use subheaders as you did for the mites.

Line 202: Add a header, may be: Mite community

Line 203-209: You are quickly focusing in detailed analyses, but you are missing the big picture. Oribatida mite abundance decreased ~4times with wild boar rooting (from 120 to 30, if I'm eyeballing the values correctly). Similarly, Mesostigmata total abundance decreased 3 times in rooting areas (remember to add the stats).

Another thing to consider in the discussion is that you measured short term effects (3 weeks after disturbance). I wonder if it gets worse over time?

Figures:

Figures are too small, I had to zoom in a lot to read numbers and labels, please provide bigger figures

Table 3 would be more appropriate as an electronic appendix

Review form: Reviewer 2 (Guillermo Bueno)

Is the manuscript scientifically sound in its present form?

Yes

Are the interpretations and conclusions justified by the results?

Yes

Is the language acceptable?

Yes

Do you have any ethical concerns with this paper?

No

Have you any concerns about statistical analyses in this paper?

Yes

Recommendation?

Major revision is needed (please make suggestions in comments)

Comments to the Author(s)

The manuscript of Dr. Maaroufi and coworkers deals with a very interesting topic, the Northern expansion of wild boar and its activities along the boreal forest. In particular, the study focused on the effect on mite population response (distinguishing between decomposers and predators) inhabiting the soils of boreal forests. While certain attention in specific habitats has been devoted to the impact of wild boar rooting to the soil organisms (fungi, earthworms, bacteria, etc.) and properties (physical and chemical properties of soils), not much attention has been focused on decomposers, besides earthworms. This is so, even despite decomposition is a key process in the nutrient cycling of ecosystems. The activity of boars turning over the soil looking for underground feeding resources has also received some attention but our understanding is local and patchy. The manuscript aims to analyze the effect of wild boar rooting affecting decomposers, in particular mites, which to my knowledge, is an understudied topic. The authors compared simulated and natural disturbances (approach used before, but not extensively) to analyze the effect of soil mixing and the removal of organic material removed by boars with their feeding activities. The experimental approach is clear and appropriate, the results are coherent and only the discussion lacks a little bit more interesting details (see the specific comments below).

Specific comments.

L126-128. Given that the study area comprises some oaks and the sampling was performed in Autumn, I wonder if the authors noticed any pattern of feeding activities around the oak acorns, a very valuable resource for boars in other areas.

L.172. Why to exclude the control? This is not very clear here, maybe adding the aim help will help.

L 176-178. Should the block random factor be applied for all models? (it sounds like it was not applied to all) Either is important or not.

What was the total number of units, 30? Please clarify. If this is so, why do you need to account for sample identity or how can this help overdispersion issues?

L188-190 "When the assumption of normality and were not met GLMM models were used with the Poisson error distribution for count data (abundance) (Crawley, 2012)" unclear, please clarify.

L190-191. "The data were log-transformed when the assumption of normality and homogeneity of variance were not met." Did the log-transformation help adjusting the model residuals to normality? In other words, were the residual normalized after that?

Discussion.

Is it possible that mite juvenile populations were lower and thus the statistical power to differentiate among disturbed and undisturbed areas were reduced? Would this potentially affect the results? If so, please add a brief text into the discussion.

I strongly suggest the authors add a couple of lines regarding the temporal aspect of the study. Differences in soil effect after rooting may depend on the time after rooting, as soil processes may be altered by different events with time (for instance snowfall, accumulation and even snow melting or plant affected decomposition). Also, it will be interesting to discuss the results in relation to the frequency of disturbance, which can be considered a proxy of disturbance intensity. For example, Bueno et al., (2013) (and references therein) found an optimum response of nitrate and ammonium at intermediate intensities of disturbance. Why do the authors think that the disturbance levels did not produce a distinctive response in the study? Please add a brief discussion of this interesting aspect.

References

Bueno, C.G., Azorín, J., Gómez-García, D. et al. Occurrence and intensity of wild boar disturbances, effects on the physical and chemical soil properties of alpine grasslands. *Plant Soil* 373, 243–256 (2013). <https://doi-org.ezproxy.utlib.ut.ee/10.1007/s11104-013-1784-z>

Decision letter (RSOS-211283.R0)

Dear Dr Maaroufi

The Editors assigned to your paper RSOS-211283 "Northward range expansion of rooting ungulates decreases soil decomposer abundances in boreal forests" have now received comments from reviewers and would like you to revise the paper in accordance with the reviewer comments and any comments from the Editors. Please note this decision does not guarantee eventual acceptance.

Please submit your revised manuscript and required files (see below) no later than 21 days from today's (ie 25-Nov-2021) date. Note: the ScholarOne system will 'lock' if submission of the

revision is attempted 21 or more days after the deadline. If you do not think you will be able to meet this deadline please contact the editorial office immediately.

on behalf of Dr Agnieszka Latawiec (Associate Editor) and Pete Smith (Subject Editor)
openscience@royalsociety.org

Associate Editor Comments to Author (Dr Agnieszka Latawiec):

Associate Editor: 1

Comments to the Author:

Dear Authors

Although interesting, your paper requires major revision to be considered in RSOS. Please address both reviewers, especially comments from the reviewer 1.

Reviewer comments to Author:

Reviewer: 1

Comments to the Author(s)

I reviewed the manuscript "Northward range expansion of rooting ungulates decreases soil decomposer abundances in boreal forest" by Maaroufi et al. In this study the authors investigated the effects of natural and simulated rooting on soil mite community. Their results show a decline in the abundance of adult mites in natural and simulated rotting areas in comparison to control areas. This study is interesting in that fewer studies have looked into the effects of wild boar rooting in the soil community. However, I found it limited in the implications. It would have been nice to see changes in the food web or ecosystem processes in response to these changes as suggested in the introduction. Measuring some other response variables such as the abundance of other members of the food web, or organic matter turnover, or soil respiration, might help to show the relevance of your findings. Otherwise your results are mainly descriptive. Also, I find the title a bit misleading, as you only measured the mite community, not the soil decomposer community.

Other suggestions:

Line 195: stats are missing

Line 197: stats are missing

In general rather than describing the significance of the statistical test, I recommend describing the biological results, e.g. Soil stoichiometry and parameters did not vary with rooting treatments

(STATS). Similarly, you could say: Total soil C, N, P and N:P ratio did not vary in response to natural wild boar rooting (Stats).

Line 195-201: It is a bit confusing to follow whether you are describing the results of the simulated rooting experiment or the natural rooting experiment, as you introduced them in a different order in the methods. I had to check the tables to understand which you were referring to. I suggest including a sentence at the beginning of this section saying that you found different effects on soil parameters depending on whether the disturbance was natural or simulated, and then describe each one. Alternatively you could use subheaders as you did for the mites.

Line 202: Add a header, may be: Mite community

Line 203-209: You are quickly focusing in detailed analyses, but you are missing the big picture. Oribatida mite abundance decreased ~4times with wild boar rooting (from 120 to 30, if I'm eyeballing the values correctly). Similarly, Mesostigmata total abundance decreased 3 times in rooting areas (remember to add the stats).

Another thing to consider in the discussion is that you measured short term effects (3 weeks after disturbance). I wonder if it gets worse over time?

Figures:

Figures are too small, I had to zoom in a lot to read numbers and labels, please provide bigger figures

Table 3 would be more appropriate as an electronic appendix

Reviewer: 2

Comments to the Author(s)

The manuscript of Dr. Maaroufi and coworkers deals with a very interesting topic, the Northern expansion of wild boar and its activities along the boreal forest. In particular, the study focused on the effect on mite population response (distinguishing between decomposers and predators) inhabiting the soils of boreal forests. While certain attention in specific habitats has been devoted to the impact of wild boar rooting to the soil organisms (fungi, earthworms, bacteria, etc.) and properties (physical and chemical properties of soils), not much attention has been focused on decomposers, besides earthworms. This is so, even despite decomposition is a key process in the nutrient cycling of ecosystems. The activity of boars turning over the soil looking for underground feeding resources has also received some attention but our understanding is local and patchy. The manuscript aims to analyze the effect of wild boar rooting affecting decomposers, in particular mites, which to my knowledge, is an understudied topic. The authors compared simulated and natural disturbances (approach used before, but not extensively) to analyze the effect of soil mixing and the removal of organic material removed by boars with their feeding activities. The experimental approach is clear and appropriate, the results are coherent and only the discussion lacks a little bit more interesting details (see the specific comments below).

Specific comments.

L126-128. Given that the study area comprises some oaks and the sampling was performed in Autumn, I wonder if the authors noticed any pattern of feeding activities around the oak acorns, a very valuable resource for boars in other areas.

L.172. Why to exclude the control? This is not very clear here, maybe adding the aim help will help.

L 176-178. Should the block random factor be applied for all models? (it sounds like it was not applied to all) Either is important or not.

What was the total number of units, 30? Please clarify. If this is so, why do you need to account for sample identity or how can this help overdispersion issues?

L188-190 “When the assumption of normality and were not met GLMM models were used with the Poisson error distribution for count data (abundance) (Crawley, 2012)” unclear, please clarify.

L190-191. “The data were log-transformed when the assumption of normality and homogeneity of variance were not met.” Did the log-transformation help adjusting the model residuals to normality? In other words, were the residual normalized after that?

Discussion.

Is it possible that mite juvenile populations were lower and thus the statistical power to differentiate among disturbed and undisturbed areas were reduced? Would this potentially affect the results? If so, please add a brief text into the discussion.

I strongly suggest the authors add a couple of lines regarding the temporal aspect of the study. Differences in soil effect after rooting may depend on the time after rooting, as soil processes may be altered by different events with time (for instance snowfall, accumulation and even snow melting or plant affected decomposition). Also, it will be interesting to discuss the results in relation to the frequency of disturbance, which can be considered a proxy of disturbance intensity. For example, Bueno et al., (2013) (and references therein) found an optimum response of nitrate and ammonium at intermediate intensities of disturbance. Why do the authors think that the disturbance levels did not produce a distinctive response in the study? Please add a brief discussion of this interesting aspect.

References

Bueno, C.G., Azorín, J., Gómez-García, D. et al. Occurrence and intensity of wild boar disturbances, effects on the physical and chemical soil properties of alpine grasslands. *Plant Soil* 373, 243–256 (2013). <https://doi-org.ezproxy.utlib.ut.ee/10.1007/s11104-013-1784-z>

===PREPARING YOUR MANUSCRIPT===

If you have been asked to revise the written English in your submission as a condition of publication, you must do so, and you are expected to provide evidence that you have received language editing support. The journal would prefer that you use a professional language editing service and provide a certificate of editing, but a signed letter from a colleague who is a fluent speaker of English is acceptable. Note the journal has arranged a number of discounts for authors using professional language editing services (<https://royalsociety.org/journals/authors/benefits/language-editing/>).

===PREPARING YOUR REVISION IN SCHOLARONE===

Author's Response to Decision Letter for (RSOS-211283.R0)

See Appendix A.

RSOS-211283.R1 (Revision)

Review form: Reviewer 1

Is the manuscript scientifically sound in its present form?

Yes

Are the interpretations and conclusions justified by the results?

Yes

Is the language acceptable?

Yes

Do you have any ethical concerns with this paper?

No

Have you any concerns about statistical analyses in this paper?

No

Recommendation?

Reject

Comments to the Author(s)

The revised version of the manuscript addressed all my suggestions. I appreciate the effort to highlight the implications of your study.

Review form: Reviewer 3

Is the manuscript scientifically sound in its present form?

Yes

Are the interpretations and conclusions justified by the results?

Yes

Is the language acceptable?

No

Do you have any ethical concerns with this paper?

No

Have you any concerns about statistical analyses in this paper?

No

Recommendation?

Accept with minor revision (please list in comments)

Comments to the Author(s)

I have reviewed the manuscript titled “Northward range expansion of rooting ungulates decreases detritivore and predatory mite abundances in boreal forests” by Maaroufi et al. The study investigates the effects of both natural and artificial rooting on mite communities and soil stoichiometry using exclosures, artificial treatments, and methods to induce wild boar rooting in a boreal study area. They found both methods of rooting to decrease mite abundance and found that wild boar rooting was associated with decreased soil C:N and increased soil C:P. The authors discuss that a decrease in mite abundance to rooting is likely due to a decrease in available mite habitat and that the increase in C:N is likely due to reduced plant activity (root removal). Overall, I think the manuscript has been effectively revised, and I welcome the changes to the discussion, particularly the discussion regarding time scale. I do have some comments regarding the methods that should help with clarity and repeatability.

Major comments:

Areas of no apparent rooting activity were chosen, but how long after rooting activity does it take for soil to reappear as normal. In the discussion, the authors state that rooting can have an effect on soil communities for up to 8 years. I imagine that during 8 years, soil can revert back to looking undisturbed. How did the authors ensure that this wasn't confounding their study? Or what are the assumptions regarding this? Given that the authors found effects from both natural and artificial rooting treatments, I suspect it's not a great issue. However, if this study were to be recreated in a study area with higher boar populations and more rooting activity, then it would be good to know how the author approached this.

I am confused over the methods in the natural wild boar rooting section. I read this section and concluded that the natural rooting plots were compared to the control plots inside exclosures, but looking at the data provided I see that the values for the control treatment within the artificial rooting tab are different than the natCrt treatment. Re-reading this section after looking at the data I now see that there were two treatments in this wild boar section. However, there is no explanation for how the control plot was set up. I assume it was also an exclosure? And both rooted and non-rooted areas were sprayed with tar? Please clarify. Also, it was unclear here that the 12 plots were split between rooting areas and non-rooted areas. I see that one previous reviewer had trouble interpreting this section. I suggest a re-write of this section. Potentially, a diagram showing the overall study design could be helpful.

Minor comments:

Line 29- as the reader, I would appreciate a very brief/simple explanation for why the study used both natural and simulated rooting in the abstract. I see in the next sentence an explanation for

the study design. Maybe stating which type of rooting represented these processes in the next sentence down (line 31) could add clarity.

Line 50- change to "several studies have provided"

Line 50- are wild boars considered large ungulates? I am not well read on wild boars so I don't know, but they seem like they would be on the smaller size of ungulates. Consider dropping the word large here to make the paragraph run more smoothly into the boar literature. It made me expect a comparison on the effects of ungulates based on body size.

Line 97- I think a better subtitle for this section would be "Site selection and enclosure design". There is no explanation of experimental treatment design in this section

Line 98- what do you define as being similar species compositions? And what was the approximate size of a site?

Lines 115 and 118- add spacing between values and units. I saw this problem throughout the manuscript as well. All cases of this should be addressed.

Line 132- is there a product brand that you can associate with this soil mixer? Something for a reader to look up and purchase themselves?

Line 138-139- I don't quite understand the timeline of this method. Was it one minute of both soil mixing and hand food removal per entire plot or per each placement of the metal frame?

Line 141- to clarify, this means there were two natural plots paired to an enclosure? If so, please clarify the wording. Also please specify how big these plots were.

Line 145 and 146- I think these should be called "sites" rather than "areas" for continuity in the Experimental Design section

Line 155- change to "We excluded one of the six natural rooting sites"

Figure 1 - I suggest making bars in panels A and D uniform given that the comparison between un-rooted and rooted areas is consistent across panels. I also suggest adding Abundance to the y-axis title in panels B and E.

Figure 2 - same comment as for figure 1, I don't see the point in color coding panels A and B. It just adds more for the eye to distinguish in the plots and the figure guide. Same comment about y-axis as for figure 1.

Decision letter (RSOS-211283.R1)

Dear Dr Maaroufi,

On behalf of the Editors, we are pleased to inform you that your Manuscript RSOS-211283.R1 "Northward range expansion of rooting ungulates decreases detritivore and predatory mite abundances in boreal forests" has been accepted for publication in Royal Society Open Science subject to minor revision in accordance with the referees' reports. Please find the referees' comments along with any feedback from the Editors below my signature.

Please submit your revised manuscript and required files (see below) no later than 7 days from today's (ie 01-Jun-2022) date. Note: the ScholarOne system will 'lock' if submission of the revision

is attempted 7 or more days after the deadline. If you do not think you will be able to meet this deadline please contact the editorial office immediately.

on behalf of Dr Agnieszka Latawiec (Associate Editor) and Pete Smith (Subject Editor)
openscience@royalsociety.org

Reviewer comments to Author:

Reviewer: 1

Comments to the Author(s)

The revised version of the manuscript addressed all my suggestions. I appreciate the effort to highlight the implications of your study.

Reviewer: 3

Comments to the Author(s)

I have reviewed the manuscript titled "Northward range expansion of rooting ungulates decreases detritivore and predatory mite abundances in boreal forests" by Maaroufi et al. The study investigates the effects of both natural and artificial rooting on mite communities and soil stoichiometry using exclosures, artificial treatments, and methods to induce wild boar rooting in a boreal study area. They found both methods of rooting to decrease mite abundance and found that wild boar rooting was associated with decreased soil C:N and increased soil C:P. The authors discuss that a decrease in mite abundance to rooting is likely due to a decrease in available mite habitat and that the increase in C:N is likely due to reduced plant activity (root removal). Overall, I think the manuscript has been effectively revised, and I welcome the changes to the discussion, particularly the discussion regarding time scale. I do have some comments regarding the methods that should help with clarity and repeatability.

Major comments:

Areas of no apparent rooting activity were chosen, but how long after rooting activity does it take for soil to reappear as normal. In the discussion, the authors state that rooting can have an effect on soil communities for up to 8 years. I imagine that during 8 years, soil can revert back to looking undisturbed. How did the authors ensure that this wasn't confounding their study? Or what are the assumptions regarding this? Given that the authors found effects from both natural and artificial rooting treatments, I suspect it's not a great issue. However, if this study were to be recreated in a study area with higher boar populations and more rooting activity, then it would be good to know how the author approached this.

I am confused over the methods in the natural wild boar rooting section. I read this section and concluded that the natural rooting plots were compared to the control plots inside exclosures, but looking at the data provided I see that the values for the control treatment within the artificial

rooting tab are different than the natCrt treatment. Re-reading this section after looking at the data I now see that there were two treatments in this wild boar section. However, there is no explanation for how the control plot was set up. I assume it was also an enclosure? And both rooted and non-rooted areas were sprayed with tar? Please clarify. Also, it was unclear here that the 12 plots were split between rooting areas and non-rooted areas. I see that one previous reviewer had trouble interpreting this section. I suggest a re-write of this section. Potentially, a diagram showing the overall study design could be helpful.

Minor comments:

Line 29- as the reader, I would appreciate a very brief/simple explanation for why the study used both natural and simulated rooting in the abstract. I see in the next sentence an explanation for the study design. Maybe stating which type of rooting represented these processes in the next sentence down (line 31) could add clarity.

Line 50- change to "several studies have provided"

Line 50- are wild boars considered large ungulates? I am not well read on wild boars so I don't know, but they seem like they would be on the smaller size of ungulates. Consider dropping the word large here to make the paragraph run more smoothly into the boar literature. It made me expect a comparison on the effects of ungulates based on body size.

Line 97- I think a better subtitle for this section would be "Site selection and enclosure design". There is no explanation of experimental treatment design in this section

Line 98- what do you define as being similar species compositions? And what was the approximate size of a site?

Lines 115 and 118- add spacing between values and units. I saw this problem throughout the manuscript as well. All cases of this should be addressed.

Line 132- is there a product brand that you can associate with this soil mixer? Something for a reader to look up and purchase themselves?

Line 138-139- I don't quite understand the timeline of this method. Was it one minute of both soil mixing and hand food removal per entire plot or per each placement of the metal frame?

Line 141- to clarify, this means there were two natural plots paired to an enclosure? If so, please clarify the wording. Also please specify how big these plots were.

Line 145 and 146- I think these should be called "sites" rather than "areas" for continuity in the Experimental Design section

Line 155- change to "We excluded one of the six natural rooting sites"

Figure 1 – I suggest making bars in panels A and D uniform given that the comparison between un-rooted and rooted areas is consistent across panels. I also suggest adding Abundance to the y-axis title in panels B and E.

Figure 2 – same comment as for figure 1, I don't see the point in color coding panels A and B. It just adds more for the eye to distinguish in the plots and the figure guide. Same comment about y-axis as for figure 1.

===PREPARING YOUR MANUSCRIPT===

one version should clearly identify all the changes that have been made (for instance, in coloured highlight, in bold text, or tracked changes);

===PREPARING YOUR REVISION IN SCHOLARONE===

- If you are providing image files for potential cover images, please upload these at this step, and inform the editorial office you have done so. You must hold the copyright to any image provided.
- A copy of your point-by-point response to referees and Editors. This will expedite the preparation of your proof.

- Ensure that your data access statement meets the requirements at <https://royalsociety.org/journals/authors/author-guidelines/#data>. You should ensure that you cite the dataset in your reference list. If you have deposited data etc in the Dryad repository, please only include the 'For publication' link at this stage. You should remove the 'For review' link.
- If you are requesting an article processing charge waiver, you must select the relevant waiver option (if requesting a discretionary waiver, the form should have been uploaded, see 'File upload' above).
- If you have uploaded any electronic supplementary (ESM) files, please ensure you follow the guidance at <https://royalsociety.org/journals/authors/author-guidelines/#supplementary-material> to include a suitable title and informative caption. An example of appropriate titling and captioning may be found at https://figshare.com/articles/Table_S2_from_Is_there_a_trade-off_between_peak_performance_and_performance_breadth_across_temperatures_for_aerobic_scope_in_teleost_fishes_/3843624.

Author's Response to Decision Letter for (RSOS-211283.R1)

See Appendix B.

Decision letter (RSOS-211283.R2)

Dear Dr Maaroufi:

I am pleased to inform you that your manuscript entitled "Northward range expansion of rooting ungulates decreases detritivore and predatory mite abundances in boreal forests" is now accepted for publication in Royal Society Open Science.

Please ensure that you send to the editorial office an editable version of your accepted manuscript, and individual files for each figure and table included in your manuscript. We note that you refer to a number of tables in your manuscript, but these are not present in the document supplied. You can send these in a zip folder if more convenient. Failure to provide these files may delay the processing of your proof.

Please remember to make any data sets or code libraries 'live' prior to publication, and update any links as needed when you receive a proof to check - for instance, from a private 'for review'

URL to a publicly accessible 'for publication' URL. It is also good practice to add data sets, code and other digital materials to your reference list.

Royal Society Open Science is a fully open access journal. A payment may be due before your article is published. Our partner Copyright Clearance Centre will contact the corresponding author about your open access options (if you have any queries regarding fees, please see <https://royalsocietypublishing.org/rsos/charges> or contact authorfees@royalsociety.org).

on behalf of Dr Agnieszka Latawiec (Associate Editor) and Professor Pete Smith (Subject Editor).

Follow Royal Society Publishing on Twitter: @RSocPublishing
Follow Royal Society Publishing on Facebook:
<https://www.facebook.com/RoyalSocietyPublishing/>
Read Royal Society Publishing's blog:
<https://royalsociety.org/blog/blogsearchpage/?category=Publishing>

Appendix A

Revised manuscript **RSOS-211283**

November 29, 2021

Dear Editor,

Thank you for the opportunity to revise our manuscript “**Northward range expansion of rooting ungulates decreases detritivore and predatory mite abundances in boreal forests**”. We appreciate the overall positive and constructive tone from both reviewers, and have made a majority of the changes they requested. Our detailed responses to their comments are specified below.

Thank you again for serving as the handling editor for our manuscript, and we look forward to a decision in due course.

Sincerely,

Dr. Nadia Maaroufi
Swedish University of Agricultural Sciences
Dept. of Forest Mycology and Plant Pathology
750 07, Uppsala, Sweden
Phone:+46 018671912

Dear Dr Maaroufi

Reviewer comments to Author:

Reviewer: 1

Comments to the Author(s)

I reviewed the manuscript “Northward range expansion of rooting ungulates decreases soil decomposer abundances in boreal forest” by Maaroufi et al. In this study the authors investigated the effects of natural and simulated rooting on soil mite community. Their results show a decline in the abundance of adult mites in natural and simulated rotting areas in comparison to control areas. This study is interesting in that fewer studies have looked into the effects of wild boar rooting in the soil community. However, I found it limited in the implications. It would have been nice to see changes in the food web or ecosystem processes in response to these changes as suggested in the introduction. Measuring some other response variables such as the abundance of other members of the food web, or organic matter turnover, or soil respiration, might help to show the relevance of your findings. Otherwise your results are mainly descriptive. Also, I find the title a bit misleading, as you only measured the mite community, not the soil decomposer community.

Response: We thank the reviewer for the positive assessment and constructive comments of our manuscript. We have now changed the title to “Northward range expansion of rooting ungulates decreases detritivore and predatory mite abundances in boreal forests”. Please see our responses to your comments.

Other suggestions:

Comment Line 195: stats are missing

Response: We have now added the stats L 199 “The soil C:N ratio significantly declined by 13% in response to natural wild boar rooting (Control 23.57 ± 1.83 and Rooting 20.18 ± 0.93 $P=0.005$; Table 1).”.

Comment Line 197: stats are missing

Response: We have now added the stats L 201-204 “Soil moisture (Control 27.20 ± 4.53 , Rooting 23.78 ± 3.22 , $P=0.017$) and organic matter content (Control 26.30 ± 8.42 , Rooting 18.49 ± 3.63 , $P=0.011$) were significantly declined under natural rooting compared to unrooted patches by 12.6% and 29.7%, respectively.”

Comment: In general rather than describing the significance of the statistical test, I recommend describing the biological results, e.g. Soil stoichiometry and parameters did not vary with rooting treatments (STATS). Similarly, you could say: Total soil C, N, P and N:P ratio did not vary in response to natural wild boar rooting (Stats).

Response: We have now added the stats throughout the all result section (L198-L234).

Line 195-201: It is a bit confusing to follow whether you are describing the results of the simulated rooting experiment or the natural rooting experiment, as you introduced them in a different order in the methods. I had to check the tables to understand which you were referring to. I suggest including a sentence at the beginning of this section saying that you found different effects on soil parameters depending on whether the disturbance was natural or simulated, and then describe each one. Alternatively you could use subheaders as you did for the mites.

Response: We have now added a main header L197 “Soil stoichiometry and parameters” and two subheaders L198 “natural rooting” and L206 “simulated rooting”. We also moved the last sentence of the paragraph to include it in the right section (L207-209).

Line 202: Add a header, may be: Mite community

Response: We have now changed the header to “Soil mite community” L211 and also to be consistent in the material and methods L147.

Line 203-209: You are quickly focusing in detailed analyses, but you are missing the big picture. Oribatida mite abundance decreased ~4times with wild boar rooting (from 120 to 30, if I’m eyeballing the values correctly). Similarly, Mesostigmata total abundance decreased 3 times in rooting areas (remember to add the stats).

Response: for clarity we have now increased the font used in the figures 1 and 2, we also added the stats, mean and standard errors L213-234, to better appreciate the variation of the samples.

Another thing to consider in the discussion is that you measured short term effects (3 weeks after disturbance). I wonder if it gets worse over time?

Response: This is a good question, we are not aware of any other studies who would have measured the effect on soil fauna. However other studies have shown that plant communities and specifically plant richness can be affecting more than 8 years after rooting occurs. We have now discussing this point in the L273-386.

Figures:

Figures are too small, I had to zoom in a lot to read numbers and labels, please provide bigger figures

Response: We have now increased the font size of all titles, axes, stats, and *post hoc* test letters.

Table 3 would be more appropriate as an electronica appendix

Response: Because reviewer 2 ask for more explanation of analysis (3) corresponding to table 3, we believe that table 3 will help the reader to understand the goals and results from this specific analysis.

Reviewer: 2

Comments to the Author(s)

The manuscript of Dr. Maaroufi and coworkers deals with a very interesting topic, the Northern expansion of wild boar and its activities along the boreal forest. In particular, the study focused on the effect on mite population response (distinguishing between decomposers and predators) inhabiting the soils of boreal forests. While certain attention in specific habitats has been devoted to the impact of wild boar rooting to the soil organisms (fungi, earthworms, bacteria, etc.) and properties (physical and chemical properties of soils), not much attention has been focused on decomposers, besides earthworms. This is so, even despite decomposition is a key process in the nutrient cycling of ecosystems. The activity of boars turning over the soil looking for underground feeding resources has also received some attention but our understanding is local and patchy. The manuscript aims to analyze the effect of wild boar rooting affecting decomposers, in particular mites, which to my knowledge, is an understudied topic. The authors compared simulated and natural disturbances (approach used before, but not extensively) to analyze the effect of soil mixing and the removal of organic material removed by boars with their feeding activities. The experimental approach is clear and appropriate, the results are coherent and only the discussion lacks a little bit more interesting details (see the specific comments below).

Response: We thank the reviewer for the positive assessment of our manuscript. Please see below our responses to your comments.

Specific comments.

L126-128. Given that the study area comprises some oaks and the sampling was performed in Autumn, I wonder if the authors noticed any pattern of feeding activities around the oak acorns, a very valuable resource for boars in other areas.

Response: this is a good point, oak trees were occurring at low density and at the time of the experiment we observed only rooting around Scots pines and Norway spruce.

L.172. Why to exclude the control? This is not very clear here, maybe adding the aim help will help.

Response: the aim of excluding the control from the third analysis aim to disentangle between the effect of bioturbation and removal as it is not possible to have a removal

treatment without disturbing the soil horizon. We have clarified the aim of the analysis (3) L75-78.

L 176-178. Should the block random factor be applied for all models? (it sounds like it was not applied to all) Either is important or not.

Response: block as a random effect was applied for all models. We have now changed the wording in the material and methods L181-182

What was the total number of units, 30? Please clarify. If this is so, why do you need to account for sample identity or how can this help overdispersion issues?

Response:

For the natural rooting experiment, we had 5 sites with 2 subsites having each 1 rooting area and an unrooted area. One sample was taken per subsites and the average was used in our analyses making a total of n=10 units.

For the simulated rooting experiment, we had 5 sites with 5 plots each, 2 samples were taken per plots and the average was used in our analyses making a total of n=25 units.

We now added the number of units in the material and methods L157-159.

We needed to account for overdispersion for our dataset and used sample identity as a random effect. Overdispersion needed to be addressed as it can bias both means and SE of parameters estimate. Thus, the model needs more information that is provided by the sample ID random effect.

Please see also *Harrison X.A. 2014. Using observation-level random effects to model overdispersion in count data in ecology and evolution. PeerJ 2:e616;DOI 10.7717/peerj.616*

L188-190 “When the assumption of normality and were not met GLMM models were used with the Poisson error distribution for count data (abundance) (Crawley, 2012)” unclear, please clarify.

Response: yes, some words were missing from the sentence, it should read When the assumption of normality and homogeneity of variance were not met GLMM models were used with the Poisson error distribution for count data (abundance) (Crawley, 2012).

L190-191. “The data were log-transformed when the assumption of normality and homogeneity of variance were not met.” Did the log-transformation help adjusting the model residuals to normality? In other words, were the residual normalized after that?

Response: It did by using a log transformation.

Discussion.

Is it possible that mite juvenile populations were lower and thus the statistical power to differentiate among disturbed and undisturbed areas were reduced? Would this potentially affect the results? If so, please add a brief text into the discussion.

Response: Statistical power is mainly influenced by sample size rather than the data itself and the sample size for adult and juveniles were the same in our study.

I strongly suggest the authors add a couple of lines regarding the temporal aspect of the study. Differences in soil effect after rooting may depend on the time after rooting, as soil processes may be altered by different events with time (for instance snowfall, accumulation and even snow melting or plant affected decomposition). Also, it will be interesting to discuss the results in relation to the frequency of disturbance, which can be considered a proxy of disturbance intensity. For example, Bueno et al., (2013) (and references therein) found an optimum response of nitrate and ammonium at intermediate intensities of disturbance. Why do the authors think that the disturbance levels did not produce a distinctive response in the study? Please add a brief discussion of this interesting aspect.

References

Bueno, C.G., Azorín, J., Gómez-García, D. et al. Occurrence and intensity of wild boar disturbances, effects on the physical and chemical soil properties of alpine grasslands. *Plant Soil* 373, 243–256 (2013). <https://doi-org.ezproxy.utlib.ut.ee/10.1007/s11104-013-1784-z>

Response: We thank the reviewer for the interesting comments, we have now added two paragraphs L273-L295 discussing temporality, intensity and frequency of wild boar rooting.

Appendix B

5th of June 2022

Dear *Royal Society Open Science* Editorial Office,

Thank you for the opportunity to revise our manuscript, titled “Northward range expansion of rooting ungulates decreases detritivore and predatory mite abundances in boreal forests”.

We hope you find our revision satisfactory, and we look for a final decision on the manuscript in due course.

Sincerely,

Nadia Maaroufi and co-authors

Reviewer comments to Author:

Reviewer: 1

Comments to the Author(s)

The revised version of the manuscript addressed all my suggestions. I appreciate the effort to highlight the implications of your study.

We thank the reviewer for the positive assessment of the manuscript.

Reviewer: 3

Comments to the Author(s)

I have reviewed the manuscript titled “Northward range expansion of rooting ungulates decreases detritivore and predatory mite abundances in boreal forests” by Maaroufi et al. The study investigates the effects of both natural and artificial rooting on mite communities and soil stoichiometry using exclosures, artificial treatments, and methods to induce wild boar rooting in a boreal study area. They found both methods of rooting to decrease mite abundance and found that wild boar rooting was associated with decreased soil C:N and increased soil C:P. The authors discuss that a decrease in mite abundance to rooting is likely due to a decrease in available mite habitat and that the increase in C:N is likely due to reduced plant activity (root removal).

Overall, I think the manuscript has been effectively revised, and I welcome the changes to the discussion, particularly the discussion regarding time scale. I do have some comments regarding the methods that should help with clarity and repeatability.

We thank the reviewer for its constructive comments.

Major comments:

Areas of no apparent rooting activity were chosen, but how long after rooting activity does it take for soil to reappear as normal. In the discussion, the authors state that rooting can have an effect on soil communities for up to 8 years. I imagine that during 8 years, soil can revert back to looking undisturbed. How did the authors ensure that this wasn't confounding their study? Or what are the assumptions regarding this? Given that the authors found effects from both natural and artificial rooting treatments, I suspect it's not a great issue. However, if this

study were to be recreated in a study area with higher boar populations and more rooting activity, then it would be good to know how the author approached this.

Response: This is a good point. We have now added clarification L102-103. At each site, areas with no apparent rooting activity were chosen. These sites had intact plant understory layer (i.e., no sign of trampling or other damages) and typical plant species composition as the rest of the surrounded forest.

I am confused over the methods in the natural wild boar rooting section. I read this section and concluded that the natural rooting plots were compared to the control plots inside enclosures, but looking at the data provided I see that the values for the control treatment within the artificial rooting tab are different than the natCrt treatment. Re-reading this section after looking at the data I now see that there were two treatments in this wild boar section. However, there is no explanation for how the control plot was set up. I assume it was also an enclosure? And both rooted and non-rooted areas were sprayed with tar? Please clarify. Also, it was unclear here that the 12 plots were split between rooting areas and non-rooted areas. I see that one previous reviewer had trouble interpreting this section. I suggest a re-write of this section. Potentially, a diagram showing the overall study design could be helpful.

Response: we thank the reviewer for rising this point. We have now added L147-149, an explanation how the non-rooted areas were selected. The non-rooted plots were paired with the adjacent wild boar rooted plots, and consisted of areas with no sign of trampling and rooting.

Minor comments:

Line 29- as the reader, I would appreciate a very brief/simple explanation for why the study used both natural and simulated rooting in the abstract. I see in the next sentence an explanation for the study design. Maybe stating which type of rooting represented these processes in the next sentence down (line 31) could add clarity.

Response: The word limit of the abstract has been reached (200 words) and we already explained the reason L31-32 “Our simulated rooting experiment aimed to disentangle the effects of a) bioturbation due to soil mixing and b) removing organic material (wild boar food resources) on soil mites”.

Line 50- change to “several studies have provided”

Response: change made

Line 50- are wild boars considered large ungulates? I am not well read on wild boars so I don't know, but they seem like they would be on the smaller size of ungulates. Consider dropping the word large here to make the paragraph run more smoothly into the boar literature. It made me expect a comparison on the effects of ungulates based on body size.

Response: change made

Line 97- I think a better subtitle for this section would be “Site selection and enclosure design”. There is no explanation of experimental treatment design in this section

Response: change made

Line 98- what do you define as being similar species compositions? And what was the approximate size of a site?

Response: we have now added the size site (0.1 ha) L98 and we have added in brackets to refer the reader to the study site section to the vegetation composition (L99).

Lines 115 and 118- add spacing between values and units. I saw this problem throughout the manuscript as well. All cases of this should be addressed.

Response: change made throughout the manuscript

Line 132- is there a product brand that you can associate with this soil mixer? Something for a reader to look up and purchase themselves?

Response: we have added the name of the soil mixer L 135.

Line 138-139- I don't quite understand the timeline of this method. Was it one minute of both soil mixing and hand food removal per entire plot or per each placement of the metal frame?

Response: we have now clarified the resource removal timing L 141“[...] food resources was removed with one hand during 1 min per frame....”

Line 141- to clarify, this means there were two natural plots paired to an enclosure? If so, please clarify the wording. Also please specify how big these plots were.

Response: we specified the area L., however we do not see any problem with the wording.

Line 145 and 146- I think these should be called “sites” rather than “areas” for continuity in the Experimental Design section

Response: change made

Line 155- change to “We excluded one of the six natural rooting sites”

Response: change made

Figure 1 – I suggest making bars in panels A and D uniform given that the comparison between un-rooted and rooted areas is consistent across panels. I also suggest adding Abundance to the y-axis title in panels B and E. Figure 2 – same comment as for figure 1, I don't see the point in color coding panels A and B. It just adds more for the eye to distinguish in the plots and the figure guide. Same comment about y-axis as for figure 1.

Response: We thank the reviewer for these suggestions. However, we note that reviewers 1 and 2 did not comment on the colour choices. We believe that the colour coding is a matter of aesthetics and that readers would appreciate the presence of colours to help interpreting the graphs. Regarding the addition of the y-axis title also for plots B and E, we think that this repetition will not improve the readability of the figures. However, we will make this change if the editor feels it is necessary.